# RETRIEVAL-AUGMENTED EDITING GENERATION: IMPACT OF KNOWLEDGE EDITING AND FINE-TUNING ON RAG

## ABSTRACT

The knowledge embedded in Large Language Models (LLMs) is static, tied to the time when the training data was collected. While Retrieval-Augmented Generation (RAG) methods are widely used to introduce new knowledge, they simply rely on retrieved information for reasoning without integrating it into the model's parameters. This limits the model's ability for long-term knowledge retention and autonomous learning. To overcome this, in this work, we propose the **R**etrieval-**A**ugmented **E**diting **G**eneration (RAEG) framework for open-domain question answering (ODQA) tasks. RAEG enhances model generation performance by first editing the retrieved paragraphs to inject necessary knowledge, followed by an augmented generation phase. This dual mechanism—combining knowledge injection and retrieval augmentation—provides complementary advantages in the reasoning process. When the injected knowledge alone is insufficient for accurate generation, the model can rely on the retrieved information to compensate, and conversely, when retrieval yields suboptimal results, the injected knowledge ensures continuity and accuracy in the response. This interplay between internalized and externally sourced knowledge reinforces the model's ability to produce correct answers, thereby enhancing overall task performance. We explore the impact of two key methods for knowledge injection: Knowledge Editing (KE) and Parameter-Efficient Fine-Tuning (PEFT), and analyze how modifying the model's parameters influences its reasoning abilities and generation outcomes. To further improve RAEG's performance, we introduce a re-ranking mechanism to optimize the integration of external knowledge and apply parameter pruning to mitigate the potential drawbacks of parameter modifications during KE. Evaluations on two authoritative ODQA benchmarks show that RAEG is able to further replace RAG as a competitive method. Our data and code will be available at https://github.com/XXX/XXX.

## 1 INTRODUCTION

Large-scale pre-trained language models (LLMs) (Radford et al., 2019; Wang & Komatsuzaki, 2022; Ouyang et al., 2022) leverage self-supervised learning on vast amounts of text to encode implicit knowledge into their parameters, enabling high-quality language generation. However, the knowledge embedded in these models is static, confined to the point in time when the training data was collected. Leveraging the language understanding and generation capabilities acquired during pre-training, large language models have achieved significant success across various practical applications. However, when confronted with more nuanced downstream tasks or unknown data knowledge, relying solely on the internal knowledge reasoning of pre-trained language models often leads to issues such as knowledge hallucination and knowledge gaps. These issues are primarily attributed to the incompleteness, biases, and static nature of the training data. For example, in the open-domain question answering (ODQA) (Voorhees & Tice, 2000) task, to address these challenges, researchers adopted RAG-based (Lewis et al., 2020) fine-tuning techniques to generate high-quality responses with the help of external knowledge specific to the current task.

RAG is a widely adopted approach for handling open-domain, knowledge-intensive tasks. It works by retrieving relevant text paragraphs and incorporating them into the generation process to assist in

Figure 1: Methods across different modes: (a) **Retrieval-Augmented Generation (RAG) mode**: Generate responses using the retrieved paragraphs. (b) **Knowledge injection mode**: Generate responses by injecting key knowledge into the model. (c) **Combined RAG and knowledge injection mode**: A dual mechanism that first perform knowledge injection, followed by that combines RAG for response generation.

answering questions. However, the model does not truly internalize this external information as its own knowledge, instead relying on real-time retrieval to augment its reasoning capabilities. Thus, knowledge editing (KE) (Zhang et al., 2024a) and parameter-efficient fine-tuning (PEFT) (Ding et al., 2023) introduce more flexible approaches that goes beyond merely relying on real-time retrieval. By directly modifying specific internal parameters, the model can internalize external knowledge. This enables not only the integration of new knowledge but also allows the model to more rapidly adapt to constantly evolving task requirements. Compared to retrieval-dependent methods, this approach offers faster and more stable responses in dynamic environments, as the model has already internalized the necessary information, eliminating the need for repeated retrieval during generation. Based on this, we propose the Retrieval-Augmented Editing Generatio (RAEG) framework, which combines a knowledge-internalized model with retrieved information. This dual mechanism leverages the immediacy of internalized knowledge while also benefiting from supplementary external retrieval, thereby improving the accuracy and consistency of the generation process.

In the course of our research, we found that adjusting model parameters may impair its performance in RAG, as illustrated in figure 1. Figure (a) illustrates a scenario where only RAG is employed, and the model is able to correctly answer the question *"Who is the current president of the US?"* rely on the retrieved paragraph. In figure (b), after applying knowledge injection via KE or PEFT, the model still answer the question correctly. However, in figure (c), when both methods are combined, the changes in model parameters may affect some of the model's prior reasoning capabilities.

In this study, we focus on the ODQA task using the llama2-7B model (Touvron et al., 2023). We explored methods of injecting new knowledge through KE and PEFT. Specifically, we extracted relevant knowledge from retrieved text paragraphs and injected it into the initial model's parameters $f_\theta$ using KE or PEFT, resulting in the edited model $f_\theta^*$. Subsequently, we combined this model with RAG to further enhance the accuracy of the generation process. We investigated the effects of KE and PEFT on the model's reasoning performance when integrated with RAG. Additionally, we introduced a paragraph re-ranking mechanism to optimize the source of edited knowledge and applied parameter pruning to mitigate the impact of knowledge editing method on the model's reasoning performance.

In summary, our contributions are as follows:

- We propose a novel Retrieval-Augmented Editing Generation (RAEG) paradigm, which internalizes retrieved knowledge into the model's parameters, reducing reliance on training data in traditional RAG systems, and offering new insights for developing more robust RAG frameworks.

- Through our experiments, we explored the impact of two knowledge injection methods—Knowledge Editing (KE) and Parameter-Efficient Fine-Tuning (PEFT)—on model reasoning performance after parameter modification in the RAG framework.

- We introduce re-ranking and parameter pruning mechanisms, which further enhance the performance of RAEG and mitigate the potential negative effects of KE on RAG.

- We demonstrate the effectiveness of the RAEG framework on ODQA tasks across two QA datasets, showing how the dual mechanism of internalized knowledge combined with retrieval can improve task performance, while discussing its potential applications and future research directions.

## 2 RELATED WORK

### 2.1 PARAMETER-EFFICIENT FINE-TUNING (PEFT)

End-to-end full fine-tuning, while being the simplest and most direct approach, becomes prohibitively expensive as the scale of pre-trained models increases. To address this, Parameter-Efficient Fine-Tuning (PEFT) (Ding et al., 2023; Lialin et al., 2023) techniques have been proposed, which aim to achieve performance comparable to full fine-tuning by adjusting only fewer parameters. PEFT methods are primarily categorized into three types: **Additive PEFT** (Zhu et al., 2021; Lei et al., 2023; Chronopoulou et al., 2023), which involves inserting adapter modules into Transformer blocks for fine-tuning; **Selective PEFT** (Sung et al., 2021; Liao et al., 2023), which selectively fine-tunes a subset of existing parameters; and **Reparameterization PEFT**, which transforms the model architecture into an equivalent form for training, such as LoRa (Hu et al., 2021), which uses low-rank matrices for adjustments. Although selecting an appropriate rank for LoRA has been a significant challenge, various derivatives of LoRA, such as DyLoRA (Valipour et al., 2023), AdaLoRA (Zhang et al., 2023), and AutoLoRA (Zhang et al., 2024b), have emerged to address this issue. These techniques enhance the efficiency of parameter tuning in large pre-trained models.

### 2.2 KNOWLEDGE EDITING (KE)

The KE methods of changing parameters mainly encompasses **Meta-learning** methods (De Cao et al., 2021; Hase et al., 2023; Mitchell et al., 2022; Tan et al., 2024), which involve training a hypernetwork to learn changes $\Delta W$ in model parameters, thus avoiding direct weight updates; and **Location-then-Edit** methods (Meng et al., 2022; 2023; Li et al., 2024; Ma et al., 2023a), which identify the locations within the model where knowledge is stored using causal traces (Meng et al., 2022), and then perform edits on those specific regions. These approaches, by directly modifying parameters of specific regions, enhance the model's capability for knowledge updating.

### 2.3 RETRIEVAL-AUGMENTED GENERATION (RAG)

RAG (Lewis et al., 2020), also known as the retrieval-reading architecture (Ma et al., 2023b), enhances language models by integrating external information through retrieval. The naive RAG method relies on basic retrieval and generation processes, but often suffers from inaccurate response (Wu et al., 2024; Xiang et al., 2024) due to irrelevant or similar information. Advanced RAG addresses these issues by optimizing indexing and query processes (Gao et al., 2023; Peng et al., 2024), and employing techniques such as re-ranking (Nogueira et al., 2020; Ju et al., 2021) and context compression (Xu et al., 2024; Cheng et al., 2024) to improve retrieval precision and response quality.

## 3 PERFORMANCE OF KE AND PEFT IN RAG

### 3.1 BACKGROUND AND MOTIVATION

#### 3.1.1 DEFINITION

**Parameter-Efficient Fine-Tuning (PEFT).** PEFT is an optimization method that freezes the majority of the model's parameters while updating only a small subset. It aims to reduce the computational burden of fine-tuning while preserving the model's original knowledge structure.

**Knowledge Editing (KE).** KE directly modifies parameters $f_{\theta^l}$ in specific layer $l$ of a model to adjust or update the embedded knowledge. This approach is typically employed to correct or introduce new facts without retraining the entire model. By locally editing the model's parameters, knowledge

editing enables the model to accurately reflect newly introduced knowledge during generation, allowing it to produce answers that incorporate the most up-to-date information.

$$f_{\theta^l}^* = \mathbf{KE}(f_{\theta^l}, \mathcal{E}), l \in L \tag{1}$$

where $\mathcal{E}$ is the new knowledge to be edited, $L$ is the set of specified editing layers.

**Retrieval-Augmented Generation (RAG).** RAG is a technique that combines retrieval and generation. Its core idea is to enhance the answering capability of generative models by retrieving relevant documents $d_{Top-k}$ from external knowledge corpus $D^{|N|} = (d_1, d_2, ..., d_N)$.

$$a = \text{argmax}_a[g(a \mid [d_{Top-k}, q])]$$

$$d_{Top-k} = \text{argtop-k}[Enc(d_i)^T \cdot Enc(q)], (d_i \in D^{|N|}) \tag{2}$$

where $g(\cdot)$ represents the generative LLM. In the RAG framework, $g(\cdot)$ is denoted as $f_\theta$ (base model), whereas in our RAEG framework, it is represented as $f_\theta^*$ (post-edited model). Here, $q$ and $a$ refer to the query and answer, respectively, and $Enc(\cdot)$ stands for the sentence encoding function used within the retrieval system.

### 3.1.2 MOTIVATION

This section primarily explores the feasibility of knowledge injection within RAG systems through knowledge editing (KE) and parameter-efficient fine-tuning (PEFT). It investigates whether targeted adjustments to the model parameters can facilitate the integration of new question-answer pairs while preserving the model's original inferential capabilities, thereby enabling accurate derivation of answers from retrieved paragraphs.

Based on the above motivation, we propose the following research questions to explore the potential impact of KE and PEFT on the model's knowledge representation, reasoning abilities, and generation quality within the RAG framework.

**RQ 1:** In the RAG framework, the model relies on externally retrieved knowledge for generation. Can KE and PEFT shift the model's reliance from external knowledge to internally embedded knowledge by injecting edited information, and how might this affect the quality and consistency of the generated output?

**RQ 2:** Performing KE and PEFT on the original pre-trained model's parameters will change the model's parameters and alter its knowledge representation. While this may enhance the model's understanding of specific knowledge, does the alteration of parameters impact the model's original capabilities, particularly its ability to rely on external knowledge for reasoning during generation? If the alteration of parameters impacts the model's original capabilities, which method—KE or PEFT—has a greater impact on the model?

### 3.2 SELF-GENERATED OF SYNTHETIC KNOWLEDGE

To ensure that LLM can accurately answer the questions, it is crucial to provide correct editing facts. We employed a prompt-based generation method in the preparation of synthetic data, a technique that involves providing examples and instructions during model generation to guide the generation of specific types of output, as shown in figure 2.

With the assistance of the large language model gpt-4o-mini (Ouyang et al., 2022), we constructed a prompt that included a clear **instruction** $T_{inst}$, detailing the objectives and requirements of the generation task, and specifying the desired style of information to be extracted from the paragraph. Additionally, we provided several **examples** $T_{exa}$ consisting of paragraph-question-answer pairs to illustrate the expected output style and format, aiding the model in understanding the desired style and content for answer generation. And then specify the **target paragraph** $T_{para}$ to generate the question-answer pairs, building the required synthetic question-answer set, which is our editing facts $\mathcal{E}(Q_{syn}, A_{syn})$. The prompt input into gpt-4o-mini is as follows. For details, please refer to table 8.

$$\mathcal{E}(Q_{syn}, A_{syn}) \leftarrow \text{gpt-4o-mini}(\mathsf{prompt})$$

$$\mathsf{prompt} = T_{inst} \oplus T_{exa} \oplus T_{para}, (T_{para} \in d_{Top-k}) \tag{3}$$

The core of this method is to guide the model in converting information from the paragraph into synthetic question-answer pairs that align with the style of the examples through effective prompts. This process significantly supports subsequent KE and PEFT.

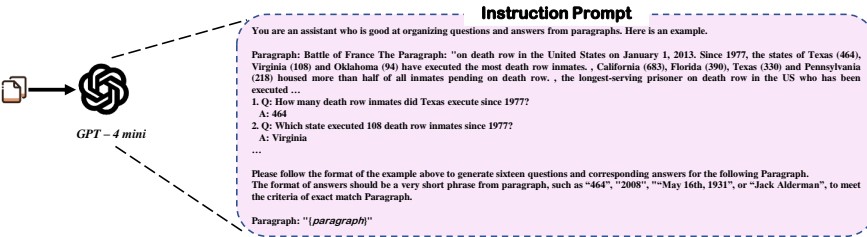

Figure 2: Self-generation of synthetic knowledge from retrieved paragraphs using prompt engineering to produce diverse and comprehensive question-answer pairs, which can be used for further knowledge injection.

## 3.3 INJECTING SYNTHETIC KNOWLEDGE INTO LLM

### 3.3.1 FOR KNOWLEDGE EDITING

We selected the MALMEN (Tan et al., 2024) method for knowledge editing, as it is more effective for editing free-text question-answer pairs and supports large-scale edits. We optimize the parameters of a specified layer using synthetic editing facts $\mathcal{E}(Q_{syn}, A_{syn})$ with the goal of maximizing the probability of the designated target $A_{syn}$. And further evaluate the capabilities of the post-edited model on test question.

Since MALMEN employs meta-learning to train a hypernetwork for editing, our objective in the context of the MALMEN method is to optimize the meta-learning hypernetwork $H$ so that it can directly generate suitable parameter adjustments $\Delta W^l$ for a given editing facts at the $l_{th}$ layer. Specifically, we aim for the hypernetwork to produce these adjustments based on the provided $(Q_{syn}, A_{syn})$, thereby improving the model's performance on new datas. To achieve this, we optimize the cross entropy loss function $\mathcal{L}_{CELF}$, the processes is defined as follows.

$$H' = \text{argmin}_H \mathcal{L}_{CELF}(-log\mathcal{P}_{(W+=H(Q_{syn}))}[A_{syn} \mid Q_{syn}])$$
$$\Delta W^l = H'(Q_{syn})$$
$$W'^l = W^l + \Delta W^l \tag{4}$$

where, $H$ denotes the initial hypernetwork, while $H'$ represents the optimized hypernetwork. The objective is to inject the $A_{syn}$ as answer of $Q_{syn}$ through adjusting $\Delta W^l$ generated in the $l$-th layer during optimization.

### 3.3.2 FOR PARAMETER-EFFICIENT FINE-TUNING

We employ Low-Rank Adaptation (LoRA) (Hu et al., 2021) as our Parameter-Efficient Fine-Tuning (PEFT) method. LoRA introduces trainable low-rank matrices into the pre-trained model, allowing us to fine-tune only a small subset of parameters while keeping the majority of the original model frozen. For the task of $Q_{syn} \rightarrow A_{syn}$ (synthetic question-to-answer mapping), LoRA allows efficient adaptation of the model by fine-tuning specific target modules. Specifically, the original parameters $W$ are kept frozen, and a small parameter matrix $\Delta W = A \cdot B$ is introduced, where $A \in \mathbb{R}^{d \times r}$ and $B \in \mathbb{R}^{r \times d}$ are both low-rank matrices. Here, $d$ represents the output dimension of the original weight matrix, and $r$ is the rank of the low-rank matrices (typically $r \ll d$). The optimization process is as follows.

$$\Delta W = \text{argmin}_{\Delta W} \mathcal{L}_{CELF}(-log\mathcal{P}_{(W+=\Delta W)}[A_{syn} \mid Q_{syn}])$$
$$W' = W + \alpha \cdot \Delta W \tag{5}$$

where $\alpha$ is a scaling factor used to balance the influence of the learned low-rank matrices with the original model weights. By using LoRA, we efficiently fine-tune the model to integrate synthetic question-answer pairs while preserving the generalization ability of the pre-trained model.

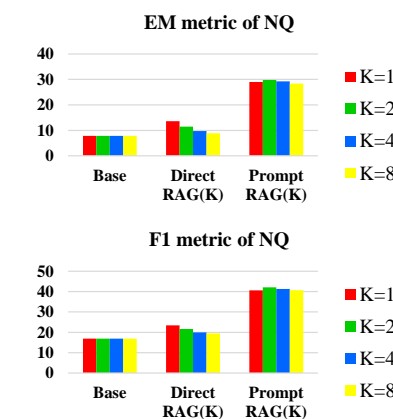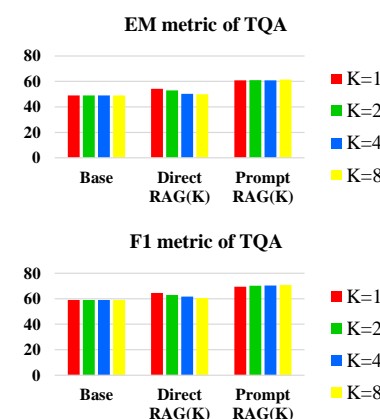

Figure 3: The impact of the number of paragraphs on RAG: This experiment explores how introducing different numbers of top-ranked paragraphs as background knowledge affects output of RAG. Using the NQ and TQA datasets, we provide top **1**, **2**, **4**, and **8** paragraphs as background knowledge to observe the quality of the generated answers.

### 3.4 EXPERIMENTAL SETUP

#### 3.4.1 METHODS

**Base Model:** For the baseline evaluation, we bypass any retrieval sources and directly input the questions into the model to observe its generated responses. The input format is shown in table 5.

**Direct-RAG(K):** In the Direct RAG setting, the top-K retrieved documents are concatenated as background knowledge and fed into the model to observe its generated responses. The input format is shown in table 6.

**Prompt-RAG(K):** To improve the accuracy of the EM metric, we followed the setup of Wang et al. (2024) by incorporating prompts into Direct-RAG, along with providing several answer examples, forming Prompt-RAG. The input format is shown in table 7.

**KE and PEFT:** KE and PEFT use the MALMEN and LoRa methods, respectively. In KE, the editing layers $L = [26, 27, 28, 29, 30, 31]$, while in PEFT, the scaling factor $\alpha$ is set to 32.

**Retriever:** we employed the Dense Passage Retrieval (DPR) (Karpukhin et al., 2020) retriever to extract relevant paragraphs from the Wikipages corpus (Vrandečić & Krötzsch, 2014).

#### 3.4.2 DATASETS AND METRICS

We utilized authoritative open-domain question-answering datasets, specifically Natural Questions (NQ) (Kwiatkowski et al., 2019) and TriviaQA (TQA) (Joshi et al., 2017), for our experiments. To assess the accuracy and quality of the responses, we employed the Extract Match (EM) and F1-Score (F1) metrics.

### 3.5 EXPERIMENTAL RESULTS AND DISCUSSION

Table 1 presents the performance results of Llama2-7B on the NQ and TQA datasets, comparing the base model, two RAG approaches, and the combined RAG methods after knowledge injection using KE and PEFT. As shown in figure 3, the model performs better when prompted with the Top-1 retrieved document. Paper Wang et al. (2024) also mentions that although retrieving more documents increases the hit rate of gold documents, irrelevant documents may introduce interference to the model. Hence, we use K=1 as the baseline.

In validating **RQ1** through our experimental results, we observed that PEFT consistently outperformed the base model across different amounts of injected paragraphs. This demonstrates that the RAEG method, constructed via PEFT, is effective in improving the performance of the RAG frame-

| Dataset | Methods | Top-1 | | Top-2 | | Top-4 | | Top-8 | |
|---|---|---|---|---|---|---|---|---|---|
| | | EM | F1 | EM | F1 | EM | F1 | EM | F1 |
| NQ | Base Model | 7.80 | 16.89 | 7.80 | 16.89 | 7.80 | 16.89 | 7.80 | 16.89 |
| | KE | 18.4 | 25.69 | 17.40 | 24.97 | 16.47 | 25.31 | 17.93 | 26.74 |
| | PEFT | 25.20 | 35.89 | 27.40 | 38.29 | 26.20 | 37.70 | 27.47 | 38.39 |
| | Direct-RAG(1) | 13.60 | 23.46 | 13.60 | 23.46 | 13.60 | 23.46 | 13.60 | 23.46 |
| | $KE_{w/\text{D-RAG}(1)}$ | 18.93 | 26.56 | 14.27 | 21.30 | 14.87 | 22.84 | 15.87 | 24.80 |
| | $PEFT_{w/\text{D-RAG}(1)}$ | 19.27 | 31.03 | 24.87 | 34.51 | 22.13 | 32.62 | 34.95 | 46.10 |
| | Prompt-RAG(1) | 29.00 | 40.61 | 29.00 | 40.61 | 29.00 | 40.61 | 29.00 | 40.61 |
| | $KE_{w/\text{P-RAG}(1)}$ | 21.53 | 29.89 | 15.27 | 22.46 | 16.93 | 25.67 | 18.0 | 26.78 |
| | $PEFT_{w/\text{P-RAG}(1)}$ | 30.13 | 41.16 | 31.93 | 43.69 | 30.2 | 42.01 | 34.07 | 44.49 |
| TQA | Base Model | 49.07 | 58.94 | 49.07 | 58.94 | 49.07 | 58.94 | 49.07 | 58.94 |
| | KE | 36.53 | 45.61 | 31.93 | 41.84 | 37.2 | 46.78 | 30.67 | 41.07 |
| | PEFT | 57.90 | 66.67 | 59.20 | 67.16 | 54.53 | 64.51 | 54.47 | 64.49 |
| | Direct-RAG(1) | 54.20 | 64.49 | 54.20 | 64.49 | 54.20 | 64.49 | 54.20 | 64.49 |
| | $KE_{w/\text{D-RAG}(1)}$ | 36.20 | 44.81 | 30.47 | 39.81 | 34.87 | 44.43 | 24.80 | 34.16 |
| | $PEFT_{w/\text{D-RAG}(1)}$ | 60.67 | 69.49 | 62.53 | 70.74 | 58.07 | 67.43 | 57.73 | 67.30 |
| | Prompt-RAG(1) | 60.8 | 69.37 | 60.8 | 69.37 | 60.8 | 69.37 | 60.8 | 69.37 |
| | $KE_{w/\text{P-RAG}(1)}$ | 37.6 | 46.33 | 31.93 | 41.04 | 37.33 | 46.96 | 26.07 | 35.90 |
| | $PEFT_{w/\text{P-RAG}(1)}$ | 62.13 | 70.51 | 61.47 | 69.87 | 56.80 | 66.06 | 56.73 | 66.09 |

Table 1: The impact of editing or fine-tuning different numbers of paragraph knowledge on RAEG. This table presents the experimental results of RAEG under varying scales of injected paragraphs. The top **1**, **2**, **4**, and **8** indicate the number of paragraphs used for knowledge injection. In the subsequent RAG stage, only the Top-1 paragraph is utilized for knowledge augmentation. D-RAG and P-RAG represent Direct RAG and Prompt RAG respectively

work by embedding edited information into the model's internal knowledge. On the other hand, the KE method showed mixed results: it surpassed the base model on the NQ dataset but did not on the TQA dataset. This suggests that the success of transferring external knowledge to internally embedded representations using KE is contingent on several factors, such as the nature of the dataset, the complexity of the questions, and the coverage of the model's pre-trained knowledge.

In investigating **RQ2**, our experiments reveal that the RAEG framework, constructed using PEFT, continues to outperform the original RAG model without negatively affecting the model's reasoning abilities. This suggests that PEFT preserves the model's capacity to leverage external knowledge for reasoning, even after modifying its internal knowledge representation. Conversely, while the KE method initially demonstrated strong performance on the NQ dataset, its performance dropped when we compared $KE_{w/\text{P-RAG}}$ with the baseline Prompt RAG. This indicates that, although KE can surpass the base model in some cases, it may compromise the model's reasoning ability as a trade-off for improving knowledge representation.

These results emphasize a critical distinction between the two methods: PEFT successfully integrates new knowledge while preserving the model's original reasoning capabilities, whereas KE, although improving specific knowledge areas, can undermine the model's reasoning performance. This highlights the need for more robust and balanced KE techniques that do not compromise original abilities of pre-trained model. In particular, developing more stable KE methods that generalize effectively across diverse datasets and handle complex reasoning tasks could significantly enhance the integration of edited knowledge into large models.

# 4 METHODOLOGY

## 4.1 IMPROVED MODULE DESIGN

### 4.1.1 RE-RANKER

Since the retrieved information is not always fully reliable, it is crucial to distinguish between useful and harmful knowledge to ensure proper alignment of the LLM's preferences. To further enhance

Figure 4: The knowledge editing process of RAEG. Relevant knowledge paragraphs are retrieved from an external corpus based on query $q$ and re-ranked by the re-ranker. The Top-K paragraphs are selected to generate synthetic knowledge for injection. To mitigate the side effects of knowledge editing, parameter pruning strategies are applied.

the accuracy of the synthesized information, we introduced a re-ranking mechanism, which refines the selection process by prioritizing the most relevant and trustworthy sources from the retrieved content.

The specific steps for training the re-ranker are shown in Algorithm 1. We construct a reranker training set from the retrieval results of the training set. For each query $q_i \in Q$ in the training set, we retrieve the top 100 documents $D_{Top}$ using a pre-trained DPR retriever (Karpukhin et al., 2020). Then, we examine each document $d_i \in D_{Top}$ to determine whether it contains the correct answer to $q_i$, labeling each document accordingly. The labels $Y_{Top}$ are binary, where $y_i$ is labeled as either '*has_answer*' or '*no_answer*'. We use the gemma-2B Gemma Team et al. (2024) model, fine-tuned by (Chen et al., 2024), as the backbone for the reranker $\mathcal{R}(\theta)$. The reranker is then fine-tuned on the constructed training subset $(q_i, d_i, y_i)$, where $d_i$ is a document retrieved for query $q_i$ and $y_i$ serves as the binary label (Yes or No) indicating whether $d_i$ contains the answer to $q_i$. The fine-tuning process optimizes the reranker using a binary cross-entropy loss function, with $y_i$ guiding the training to improve the reranker's ability to distinguish between relevant and irrelevant documents. For detailed results on the retrieval accuracy after re-ranking, please refer to appendix B.

### 4.1.2 PARAMETER PRUNING

After applying knowledge editing, we obtain the parameter update matrix $\Delta W$. Drawing inspiration from the findings of Gu et al. (2024), which suggest that smaller values in $\Delta W$ may carry less substantial editing information but can still affect the model's performance, we apply a pruning strategy. By pruning $\Delta W$, we aim to filter out parameters with smaller magnitudes, as they might contribute less to the desired knowledge update. In addition, we propose a random pruning strategy. We explore two pruning strategies to optimize the edited parameters $\Delta W$ in order to mitigate the potential side effects of KE.

---

**Algorithm 1** Re-ranker Training

---

**Requested**: Query set in training set $Q$, Retriever $DPR$, Retrieved results $(D, Y) = (d_k, y_k)_{k=1}^{100}$, Re-ranker model $R_\theta$

1: **for** each query $q_i \in Q$ **do**
2: $\quad (D, Y)_i = DPR(q_i)$
3: $\quad$ **for** each $(d_k, y_k)_{k=1}^{100}$ **do**
4: $\quad\quad$ relevance_score = $R_\theta(q_i, d_i)$
5: $\quad\quad$ loss = BCELF(relevance_score, $y_i$)   *//BCELF means binary cross-entropy loss function*
6: $\quad\quad$ Update $R_\theta$
7: $\quad$ **end for**
8: **end for**
9: Return Trained Re-ranker $R_\theta$

---

| Dataset | Methods | Top-1 | | Top-2 | | Top-4 | | Top-8 | |
|---|---|---|---|---|---|---|---|---|---|
| | | EM | F1 | EM | F1 | EM | F1 | EM | F1 |
| NQ | $KE_{w/D\text{-}RAG(1)}$ | 27.13 | 35.98 | 21.47 | 31.12 | 20.8 | 31.11 | 21.20 | 31.97 |
| | | 8.20 | 9.42 | 7.20 | 9.82 | 5.93 | 8.27 | 5.33 | 7.17 |
| | $PEFT_{w/D\text{-}RAG(1)}$ | 27.93 | 38.25 | 34.67 | 46.20 | 35.10 | 46.88 | 34.93 | 46.53 |
| | | 8.66 | 7.22 | 9.80 | 11.69 | 12.97 | 14.26 | -0.02 | 0.43 |
| | $KE_{w/P\text{-}RAG(1)}$ | 32.40 | 41.91 | 25.53 | 34.54 | 23.60 | 34.19 | 23.13 | 33.46 |
| | | 10.87 | 12.02 | 10.26 | 12.08 | 6.67 | 8.52 | 5.13 | 6.68 |
| | $PEFT_{w/P\text{-}RAG(1)}$ | 32.13 | 43.51 | 33.93 | 45.51 | 34.89 | 46.11 | 34.4 | 46.11 |
| | | 2.00 | 2.35 | 2.00 | 1.82 | 4.69 | 4.10 | 0.33 | 1.62 |
| TQA | $KE_{w/D\text{-}RAG(1)}$ | 47.60 | 58.39 | 44.20 | 55.43 | 49.53 | 60.11 | 39.07 | 50.75 |
| | | 11.40 | 13.58 | 13.73 | 15.62 | 14.66 | 15.68 | 14.27 | 16.59 |
| | $PEFT_{w/D\text{-}RAG(1)}$ | 61.40 | 70.74 | 63.47 | 72.11 | 60.27 | 70.33 | 60.53 | 69.80 |
| | | 0.73 | 1.25 | 0.94 | 1.37 | 2.20 | 2.90 | 2.80 | 2.50 |
| | $KE_{w/P\text{-}RAG(1)}$ | 50.00 | 60.24 | 46.40 | 56.87 | 52.07 | 62.10 | 40.33 | 50.46 |
| | | 12.4 | 13.91 | 14.47 | 15.83 | 14.74 | 15.14 | 14.26 | 14.56 |
| | $PEFT_{w/P\text{-}RAG(1)}$ | 62.60 | 70.88 | 63.67 | 71.67 | 59.20 | 69.08 | 60.40 | 69.61 |
| | | 0.47 | 0.37 | 2.20 | 1.80 | 2.40 | 3.02 | 3.67 | 3.52 |

Table 2: Results after re-ranking and parameter pruning. Red values indicate improvements compared to those in table 1. The knowledge editing (KE) results are based on magnitude-based pruning applied at a 30% pruning ratio.

**Random Pruning:** This strategy is inspired by the dropout mechanism, where parameters are randomly zeroed out. This helps prevent overfitting to the edited knowledge by ensuring that the model does not become overly reliant on specific edited parameters.

**Magnitude-based Pruning:** In this approach, we filter out the smallest K% of the parameter values, under the assumption that these smaller values contain less critical editing information. By zeroing out these parameters, we aim to preserve the model's performance by minimizing the difference between the edited model and the original model. The pruning operation can be formalized as follows:

$$\Delta W = \begin{cases} 0, & \text{if } \Delta W \leq \text{threshold}(K\%) \\ \Delta W, & \text{otherwise} \end{cases} \tag{6}$$

Through this dual pruning strategy, we balance the retention of critical knowledge with maintaining the integrity of the pre-existing model structure, effectively mitigating the risk of excessive edits.

## 4.2 Ablation Study of Parameter Pruning

We conducted a comprehensive study on various pruning scales using two different pruning strategies, the results as shown in table 3. The results indicate that, for both pruning strategies, performance improves progressively as the pruning ratio increases. This suggests that in extreme pruning scenarios, where a significant portion of the parameters are removed, the remaining parameters are sufficiently robust to sustain overall model performance.

At pruning scales below 50%, magnitude-based pruning significantly outperforms random pruning. This highlights the efficacy of structured pruning strategies, which rank and remove less important parameters based on their magnitudes, thus preserving the critical parameters that contribute most to the model's performance. In contrast, random pruning at low pruning ratios tends to remove key parameters indiscriminately, leading to noticeable performance degradation. At pruning scales above 50%, random pruning shows marked improvement. This suggests that while random pruning may remove some redundant parameters at higher pruning ratios, its inherent randomness still results in less stability and consistency compared to magnitude-based pruning, which remains more reliable across different scales.

These findings indicate that magnitude-based pruning is a more stable and effective approach, especially at lower pruning ratios, as it preserves model performance more effectively. However, despite

| Pruning strategy | Pruning scale | NQ | | TQA | |
|---|---|---|---|---|---|
| | | EM | F1 | EM | F1 |
| Magnitude | 10% | 32.00 | 41.51 | 48.53 | 58.88 |
| | 30% | 32.40 | 41.91 | 50.00 | 60.24 |
| | 50% | 32.53 | 42.13 | 50.73 | 60.64 |
| | 70% | 32.47 | 42.20 | 51.00 | 61.14 |
| | 90% | 33.20 | 42.95 | 52.87 | 62.73 |
| Random | 10% | 29.07 | 38.10 | 44.13 | 54.84 |
| | 30% | 29.53 | 38.99 | 47.27 | 57.67 |
| | 50% | 32.07 | 41.68 | 51.47 | 61.61 |
| | 70% | 32.73 | 43.09 | 55.87 | 65.50 |
| | 90% | 32.20 | 43.04 | 61.20 | 69.97 |

Table 3: Ablation experiment on parameter pruning: Results of two pruning strategies at different pruning scales.

its variability, random pruning demonstrates potential at higher pruning scales, particularly in extreme pruning conditions where it may still offer some practical applications.

### 4.3 FURTHER EXPERIMENTAL RESULTS AND DISCUSSION

The experimental setup in this section is identical to that in section § 3.4, ensuring a fair assessment of the improvements introduced by our designed modules.

Table 2 illustrates the further performance improvements achieved by integrating the improved modules, compared to the results in table 1.

Through the re-ranking process, we further refine the selection of documents that are more likely to contain correct answers. The information from these selected documents is then edited into the model to enhance its knowledge representation and accuracy. Parameter pruning mitigated the impact of the edited parameters on the original model's weights, resulting in an 8% to 12% performance improvement for the KE method combined with RAG across both datasets.

## 5 CONCLUSION AND FUTURE WORK

In this work, we introduced a novel framework, Retrieval-Augmented Editing Generation (RAEG), which combines knowledge injection from retrieved documents with RAG to enhance the accuracy of answer generation. The dual strategy of first injecting knowledge and then performing retrieval-augmented generation significantly improves model performance.

A key contribution of our study is the investigation of the impact of two knowledge injection techniques—Knowledge Editing (KE) and Parameter-Efficient Fine-Tuning (PEFT)—on model reasoning abilities after parameter modification. Our experiments show that while KE effectively internalizes new knowledge, it severely disrupts the model's prior reasoning capabilities. In contrast, PEFT, which operates through global fine-tuning, preserves the model's overall performance more effectively and achieves better results on open-domain question-answering (ODQA) tasks. In finally, we further enhanced RAEG by introducing a re-ranking mechanism to refine the selection of reliable knowledge sources and by employing parameter pruning to mitigate the negative effects of KE on model performance.

Future research should continue to explore and discuss the impacts of various editing techniques on parameter updates of pre-trained language models across a broader range of NLP tasks. Additionally, developing more robust components to counteract the unintended side effects of model editing and its ripple effects (Cohen et al., 2024) remains a critical challenge. Our data and code have been made available to the community to support further advances in this research direction.

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

## A    Environment Setting

All data construction, knowledge editing, and evaluation experiments were conducted on worksstations equipped with NVIDIA RTX A6000 GPUs. The initial weights of the LLama-2 (Touvron et al., 2023) language models were sourced from HuggingFace Transformers (Wolf et al., 2019), and the experiments utilized PyTorch version 2.4.0 (Paszke et al., 2019).

## B    Result of Re-ranker

Table 4 presents the retrieval accuracy results of our trained re-ranker model, comparing the performance before and after its implementation on two datasets: NQ and TQA. The results show a significant increase in retrieval accuracy across 4 top-rank type.

Higher retrieval accuracy directly affects the effectiveness of the generated synthetic knowledge, significantly enhancing the quality and reliability of the responses. Therefore, the implementation of the re-ranker not only optimizes the retrieval process but also greatly enriches the knowledge base relied upon for generating responses. This improvement enables the model to generate synthetic knowledge more effectively, increasing the accuracy and effectiveness of the final responses in the RAEG framework. Our trained re-ranker model will be released along with the associated code.

| Datasets | Top-1 Before | Top-1 After | Top-2 Before | Top-2 After | Top-4 Before | Top-4 After | Top-8 Before | Top-8 After |
|---|---|---|---|---|---|---|---|---|
| NQ | 44.60 | 62.66 | 55.73 | 69.47 | 64.47 | 76.67 | 72.93 | 80.33 |
| TQA | 56.53 | 76.27 | 65.27 | 79.67 | 72.07 | 82.53 | 76.73 | 84.47 |

Table 4: Comparison of retrieval accuracy results before and after using the re-ranker.

## C    Prompt format

This section provides a detailed overview of the input formats utilized in this paper. It includes the input format for the Base model in table 5, as well as the input formats for Direct-RAG(K) and Prompt-RAG(K) in table 6 7. Additionally, we outline the prompt templates used for self-generating 8 synthetic knowledge.

---

**Input Format of Base**

Question: {*question*}
Answer:

---

Table 5: Input Format of Base.

---

**Input Format of Direct-RAG(K)**

Knowledge:
{*Top-1 paragraph*}
...
{*Top-K paragraph*}

Question: {*question*}
Answer:

---

Table 6: Input Format of Direct-RAG(K).

**Input Format of Prompt-RAG(K)**

Knowledge:
{***Top-1 paragraph***}
...
{***Top-K paragraph***}

Base above knowledge, answer the following question with a very short phrase, such as "1998", "May 16th, 1931", or "James Bond", to meet the criteria of exact match datasets.
Question: {***question***}
Answer:

Table 7: Input Format of Prompt-RAG(K).

---

**Prompt templates for self-generating synthetic knowledge**

You are an assistant who is good at organizing questions and answers from paragraphs. Here is an example.

Paragraph: "on death row in the United States on January 1, 2013. Since 1977, the states of Texas (464), Virginia (108) and Oklahoma (94) have executed the most death row inmates. , California (683), Florida (390), Texas (330) and Pennsylvania (218) housed more than half of all inmates pending on death row. , the longest-serving prisoner on death row in the US who has been executed was Jack Alderman who served over 33 years. He was executed in Georgia in 2008. However, Alderman only holds the distinction of being the longest-serving ëxecutedïnmate so far. A Florida inmate, Gary Alvord, arrived"
1. Q: How many death row inmates did Texas execute since 1977?
A: 464
2. Q: Which state executed 108 death row inmates since 1977?
A: Virginia
3. Q: How many death row inmates did Oklahoma execute since 1977?
A: 94
4. Q: Which state housed 683 death row inmates as of January 1, 2013?
A: California
5. Q: How many inmates did Florida house on death row?
A: 390
6. Q: How many death row inmates did Texas have pending?
A: 330
7. Q: How many death row inmates did Pennsylvania house?
A: 218
8. Q: Who was the longest-serving prisoner on death row who was executed?
A: Jack Alderman
9. Q: How many years did Jack Alderman serve on death row?
A: over 33 years
10. Q: In which year was Jack Alderman executed?
A: 2008
11. Q: Which state executed Jack Alderman?
A: Georgia
12. Q: Who is noted as the longest-serving "executed" inmate?
A: Jack Alderman
13. Q: Which inmate arrived in Florida?
A: Gary Alvord
14. Q: What is the date referenced for death row statistics in the passage?
A: January 1, 2013
15. Q: Since when has the execution data been tracked in this passage?
A: 1977
16. Q: What constitutes more than half of all inmates pending on death row?
A: California, Florida, Texas, and Pennsylvania

Please follow the format of the example above to generate sixteen questions and corresponding answers for the following Paragraph. The format of answers should be a very short phrase from paragraph, such as "464", "2008", "May 16th, 1931", or "Jack Alderman", to meet the criteria of exact match Paragraph.

Paragraph: "{*paragraph*}"

---

Table 8: The prompt template format used to generate synthetic knowledge, which includes a clear instruction outlining our requirements and an example of a paragraph along with its corresponding question-answer pairs. The model is then expected to generate similar question-answer pairs for new paragraphs based on this format.

