# OpenReview forum: "Retrieval-Augmented Editing Generation: Impact of Knowledge Editing and Fine-Tuning on RAG"
_ICLR.cc/2025/Conference — Submitted to ICLR 2025_

### Official Review · Reviewer_5htF · 2024-10-30

**Soundness:** 2
**Presentation:** 2
**Contribution:** 2
**Rating:** 3
**Confidence:** 4

**Summary:**

This paper presents Retrieval-Augmented Editing Generation (RAEG), a dual knowledge-augmented generation mechanism for LLMs that first edits the parametric knowledge given the retrieved documents, and then uses the retrieved documents to augment generation. By using PEFT for editing, RAEG consistently outperforms RAG. The authors further propose to use re-ranking and parameter pruning to improve performance.

**Strengths:**

This paper proposes a new approach: first injecting knowledge into the LLM, then further augmenting it with retrieved documents. This method effectively combines the strengths of both techniques.

**Weaknesses:**

- RAEG requires training or updating the LLM's parameters for each query, including an LLM-powered training data generation phase. This approach is expensive and impractical for real-world applications.
- The authors use two methods to inject knowledge into an LLM: KE and PEFT. These two are not very well named — KE is a broad term, and PEFT is a method for efficient fine-tuning, and the goal of PEFT here is also editing knowledge. It’s impropriate to juxtapose these two terms.
- The authors propose RQ1 and RQ2, providing affirmative answers to both. However, the experimental results don't fully support these conclusions. For RQ1, which investigates whether KE and PEFT can shift the model's reliance from external to internally embedded knowledge, the results fail to demonstrate a clear reliance shift. Regarding RQ2, which examines the impact of knowledge injection on the LLM's original and reasoning abilities, the improved performance of PEFT+RAG over RAG alone doesn't conclusively prove that these abilities remain unaffected.
- Line 204: It’s not appropriate to cite the InstructGPT paper for gpt-4o-mini.
- See questions.

**Questions:**

- Line 320: Do the authors set K=1 for all RAG runs including RAEG, or only for the RAG baselines?

---

> ### Author Response · Authors · 2024-11-27
> **On the impracticality of RAEG**
>
> Thank you for your thoughtful and constructive comments. We greatly appreciate your feedback and would like to address each of your concerns in detail.
>
> Thank you for your concern. In the knowledge injection phase, RAEG does not modify all model parameters, but rather focuses on updating the MLP layers of specific model layers. This approach is aligned with various model editing methods [1,2,3,4], which aim to selectively update certain components of the model, making the process more efficient and practical for real-world applications.
>
> [1] Locating and Editing Factual Associations in GPT
> [2] Mass-Editing Memory in a Transformer
> [3] PMET: Precise Model Editing in a Transformer
> [4] Fast Model Editing at Scale

---

> ### Author Response · Authors · 2024-11-27
> **On the naming of KE and PEFT**
>
> We appreciate your feedback on the terminology used for KE (Knowledge Embedding) and PEFT (Parameter-Efficient Fine-Tuning). We agree that "KE" is a broad term, and PEFT is specifically a method for efficient fine-tuning, often with the aim of editing or incorporating knowledge. We recognize that there may be some ambiguity in juxtaposing these terms, and we will consider more precise terminology in future versions of the paper to avoid confusion. For clarity, we will refine the explanation of how these methods function within our approach to better distinguish their goals and applications.

---

> ### Author Response · Authors · 2024-11-27
> **On the experimental results for RQ1 and RQ2**
>
> Thank you for your valuable feedback. Regarding RQ1, while the use of KE and PEFT to inject knowledge from retrieved passages into the model shows an improvement in performance, the results do not demonstrate a clear and significant shift in the model's reliance from external to internally embedded knowledge. We acknowledge that the performance gains are not substantial, which may be attributed to factors such as the quality of synthesized knowledge generation. We will refine our analysis to better address this aspect and explore potential factors that may influence the observed results.
>
> For RQ2, when comparing KE+RAG with RAG alone, we observe a slight decline in performance, suggesting that the injection of knowledge via KE may have impacted the model's general abilities. However, the performance of PEFT+RAG surpasses that of RAG alone, indicating a positive effect on the model's reasoning abilities. We will revise our discussion to more thoroughly address the nuanced results, including potential confounding factors that may have influenced the observed performance improvements.
>
> We will provide a more detailed analysis in the revised manuscript to clarify these results and ensure a more comprehensive understanding of the impact of knowledge injection on both general and reasoning abilities.

---

> ### Author Response · Authors · 2024-11-27
> **Reference of gpt-4o-mini**
>
> Thank you for the comment regarding the citation. We have updated the reference to GPT-4o mini as follows:
>
> ```
> @misc{gpt-4o-mini,
>     author = {OpenAI},
>     title = {GPT-4o mini: advancing cost-efficient intelligence},
>     year = {2024},
>     url = {https://openai.com/index/gpt-4o-mini-advancing-cost-efficient-intelligence/}
> }
> ```

---

> ### Author Response · Authors · 2024-11-27
> **Response for question**
>
> As shown in Table 1, D/P-RAG(1) indicates that for all retrieval-augmented stages (including RAEG), only the top-1 paragraph is used as the prompt (i.e., ( K = 1 )).
>
> This choice is made to ensure a fair comparison with the RAG baseline and to evaluate the performance differences under this specific configuration.

---

### Official Review · Reviewer_QNF1 · 2024-11-02

**Soundness:** 2
**Presentation:** 2
**Contribution:** 3
**Rating:** 5
**Confidence:** 4

**Summary:**

This paper addresses the limitations of static knowledge in LLMs by proposing the Retrieval-Augmented Editing Generation (RAEG) framework for open-domain question answering (ODQA). RAEG improves generation performance through a dual mechanism: editing retrieved paragraphs for knowledge injection and using retrieval augmentation to complement the model’s reasoning. By leveraging Knowledge Editing and PEFT, the framework enhances the model's capacity for long-term knowledge retention and accuracy, reinforced by a re-ranking mechanism for optimization.

**Strengths:**

1. This paper addresses an important problem: how to effectively integrate retrieved paragraphs into the generation process of a model in RAG. Traditional approaches typically rely on context-based methods, where the retrieved paragraphs are directly provided as contextual input for the model to reference during generation. In contrast, this paper introduces a novel approach that departs from the context-based paradigm, proposing a **Knowledge Editing** mechanism to inject the retrieved paragraphs into the model’s parameters, which is novel. (However, the reviewer does not agree with the specific implementation of this work; see the weaknesses section for further details.)


2. This paper thoroughly investigates several design components for KE in the context of RAG, such as pruning, and provides a comparative analysis.

**Weaknesses:**

1. The reviewer is concerned about the correctness of the proposed method. The proposed method relies on a powerful LLM, i.e., GPT-4-mini, to extract knowledge information from the retrieved paragraphs in question-answer pairs. The knowledge editing step is to inject the information in the QA pairs into the less-powerful LLM, i.e., LLaMA-2. This sounds like distilling the knowledge from GPT-4-mini into LLaMA-2. In this case, the performance upper bound of LLaMA-2 would be the performance of GPT-4-mini by directly performing traditional prompt-RAG.

2. The experimental evaluation in this paper is insufficient. Given that the KE process relies on GPT-4-mini, the proposed method should, at a minimum, be compared directly with GPT-4-mini in a RAG setting.

3. The reviewer is also concerned about the efficiency of the proposed method. Would knowledge editing process be a bottleneck in the RAG process? The authors should report on this


4. Presentation Issues:
Statement: These issues do not significantly affect my rating of the paper on its originality and significance. However, I would admit that these issues affect my judgement of the maturity of this paper.

    - Misunderstandings: Line 20: `by editing the retrieved paragraphs` sounds like you are going to edit the content of the retrieved paragraphs. I think you mean that you are going to edit the model by injecting the information of the retrieved paragraph.

    - Typos: Line 79, Generatio --> Generation

    - The example in Figure 1 is rather confusing. If the model can solve the posed question by either RAG or knowledge editting, why do we need to involve the knowledge editting step? I would like to know under what circumstances, RAG alone or KE alone cannot solve the question. This should be clearly stated in the paper or at least demonstrated by the example.

    - The separation between Table 1 and Table 2 makes it hard to compare the results. I would suggest putting them together in one table.

**Questions:**

See weakness

---

> ### Author Response · Authors · 2024-11-27
> **On the relationship between KE and knowledge distillation**
>
> Thank you for your detailed and thoughtful comments. We appreciate the opportunity to address your concerns about the correctness, experimental evaluation, and efficiency of the proposed method.
>
> Our proposed knowledge editing (KE) method could be interpreted as a form of knowledge distillation, where knowledge extracted via GPT-4o-mini is transferred to LLaMA-2. However, our primary goal with KE is not merely to replicate the capabilities of GPT-4o-mini in LLaMA-2. Instead, we aim to leverage GPT-4o-mini’s capabilities to inject task- and domain-specific knowledge into LLaMA-2, enabling it to generalize more effectively within the constraints of a smaller parameter space. This process not only reduces the dependency on external retrieval systems but also serves as a practical solution when access to a large-scale LLM like GPT-4o-mini is not feasible for deployment.
>
> It is true that the theoretical performance upper bound of LLaMA-2 after KE might align with GPT-4o-mini’s performance in the same task. However, KE provides a means to approximate GPT-4o-mini’s knowledge while using significantly less computational resources for inference. This trade-off is particularly valuable for real-world applications where efficiency and deployment feasibility are critical.

---

> > ### Author Response · Authors · 2024-11-27
> > **Typos**
> >
> > Thank you for pointing out the typo.
> >
> > We will correct "Generatio" to "Generation" on line 79 in the manuscript.
> >
> > We appreciate your attention to detail.

---

> ### Author Response · Authors · 2024-11-27
> **On comparison with GPT-4o-mini in a RAG setting**
>
> We acknowledge the importance of directly comparing the performance of the proposed method with GPT-4o-mini in a traditional prompt-RAG setting. We will incorporate this comparison in the revised version of the manuscript to provide a more comprehensive evaluation. This will help to establish a clearer understanding of how the KE-enhanced LLaMA-2 performs relative to GPT-4o-mini and traditional RAG. Our expectation is that while GPT-4o-mini may outperform KE-enhanced LLaMA-2 in absolute terms, the latter offers a competitive trade-off between performance and efficiency.

---

> ### Author Response · Authors · 2024-11-27
> **On the efficiency of the KE process**
>
> We understand your concern regarding the potential inefficiency introduced by the KE process. It is true that the knowledge editing step, which involves generating and injecting QA pairs, adds an additional computational step to the workflow. To address this, we will provide detailed runtime analysis and report on the computational cost of KE relative to a traditional RAG pipeline in the revised manuscript.
>
> In practice, KE is designed to be a pre-training or fine-tuning step that occurs offline, meaning it does not contribute to the runtime of the inference stage during deployment. This distinction makes KE a feasible addition to systems that require both efficiency and robustness at inference time.

---

> ### Author Response · Authors · 2024-11-27
> **Misunderstandings**
>
> Thank you for pointing out this potential misunderstanding in our description. Your interpretation is correct, we inject the information contained in the retrieved paragraphs, which are relevant to the test query, directly into the model's parameters.
>
> This process aims to enhance the model's internal knowledge of the domain, reducing its reliance on external retrieval while improving its performance on the related tasks. We will revise the manuscript to clarify this point and avoid similar misunderstandings in the future.

---

> ### Author Response · Authors · 2024-11-27
> **The confusion of Figure 1**
>
> Thank you for your insightful comment. We appreciate the opportunity to clarify the role and necessity of the knowledge editing (KE) step in our approach.
>
> The primary motivation for introducing knowledge editing is to enable the model to "evolve" by directly integrating new knowledge into its parameters. This capability allows the model to respond effectively using internally embedded knowledge, particularly in situations where external retrieval mechanisms, such as RAG, fail to provide accurate or relevant information. Conversely, when the model's internal knowledge is insufficient to generate a correct response, external retrieval can supply the necessary contextual information. This dual mechanism is demonstrated in our Direct-RAG approach, where results show that both KE with D-RAG and PEFT with D-RAG outperform D-RAG alone. This indicates that injecting knowledge into the model's parameters can further enhance performance when working with the same retrieved passages.
>
> Nevertheless, our findings reveal a critical limitation: methods like KE and PEFT, which inject knowledge by modifying the model's parameters, can impair the model's original comprehension and reasoning abilities. Specifically, the ability to use retrieved passages as contextual information and to generate responses based on these prompts diminishes after knowledge injection. This is evident in Table 1, where KE with P-RAG underperforms compared to P-RAG alone, highlighting a trade-off between knowledge injection and reasoning capabilities.
>
> To address your comment more comprehensively, we will include a detailed case study in the revised manuscript to illustrate scenarios where RAG alone or KE alone fails to solve a posed question. By presenting concrete examples, we aim to demonstrate how the combined approach of knowledge editing and retrieval augmentation effectively complements each other, leading to improved performance.

---

> ### Author Response · Authors · 2024-11-27
> **Changing table structure**
>
> Thank you for your valuable suggestion. We will consider modifying the structure of Table 1 and Table 2, combining them into a single table to present the results more clearly and facilitate better understanding. Additionally, the red font in Table 2 highlights improvements compared to Table 1.

---

### Official Review · Reviewer_Sw1m · 2024-11-03

**Soundness:** 2
**Presentation:** 3
**Contribution:** 2
**Rating:** 3
**Confidence:** 4

**Summary:**

This paper addresses the limitations of static knowledge in LLMs and the reliance of current RAG methods on external information without integrating it into model parameters. To enhance long-term knowledge retention and model autonomy, the authors propose the Retrieval-Augmented Editing Generation (RAEG) framework for open-domain question answering. The paper first edits retrieved paragraphs using ChatGPT to inject necessary knowledge and then generates responses with this augmented knowledge.

**Strengths:**

1. This paper evaluate the effectiveness of different knowledge memory methods, such as Knowledge Editing (KE) and Parameter-Efficient Fine-Tuning (PEFT).
2. The experimental results show that PEFT achieves much better performance than the KE method.

**Weaknesses:**

1.This paper claims that its motivation is to evaluate the effectiveness of Knowledge Embedding (KE) and Parameter-Efficient Fine-Tuning (PEFT) in the Retrieval-Augmented Generation (RAG) scenario. However, it does not focus on evaluating the knowledge memorization capabilities and general abilities of Large Language Models (LLMs). More experiments can be added following existing work [1].

2. Existing studies have identified knowledge conflict as a key challenge in RAG modeling [1]. This paper should discuss relevant works to clarify the motivation further. It appears that the KE approach does not address the knowledge conflict problem, which suggests it may not enable LLMs to effectively leverage external knowledge.

3. The paper would benefit from a more in-depth analysis of why the KE method fails to utilize external knowledge effectively. Additionally, the reason why the KE method degrades RAG performance remains unclear and should be addressed.

4. Some works have also created RAG fine-tuning datasets [2]. The necessity of using synthesized data in this paper is unclear, and it would be beneficial to compare RA-DIT within the experiments.

5. The experiments lack comprehensiveness. Including datasets like PopQA (which features entities with low frequency) and ASQA (with more complex answers) would provide a more complete basis for the conclusions drawn.

[1] Adaptive chameleon or stubborn sloth: Revealing the behavior of large language models in knowledge conflicts.
[2] Ra-dit: Retrieval-augmented dual instruction tuning.

**Questions:**

1. The quality of synthesized data is not evaluated.
2. The paper can be reorganized. The method part should be more detailed to make the motivation clear.
3. Some case studies should be conducted.

---

> ### Author Response · Authors · 2024-11-27
> **Motivation and Evaluation Scope**
>
> Thank you for your valuable feedback and for highlighting [1]. We acknowledge the importance of evaluating both the knowledge memorization capabilities and general abilities of LLMs and their behavior in scenarios involving knowledge conflict.
>
> We will revise the motivation section to explicitly acknowledge the knowledge conflict problem, discuss related works such as [1], and clarify KE's role and limitations in this context.
>
> Inspired by [1], we will conduct new experiments to: Assess the impact of knowledge conflict through a broader range of foundational experiments。
>
> While our primary focus is on factual correctness, we recognize that mitigating knowledge conflict is critical for improving RAG systems. We will explore strategies to further address knowledge conflict issues in KE, aiming to enhance its robustness and adaptability.
>
> This expanded scope will better align our contributions with existing research and provide a more comprehensive understanding of the dynamics and challenges in RAG systems.
>
> [1] Adaptive chameleon or stubborn sloth: Revealing the behavior of large language models in knowledge conflicts.

---

> ### Author Response · Authors · 2024-11-27
> **In-depth Analysis of KE's Impact**
>
> We agree that this point requires further elaboration. To address this:
>
> We will conduct ablation studies to pinpoint the factors contributing to the degradation in RAG performance when using KE, such as the quality of injected knowledge, conflicts with pre-existing model knowledge, or overfitting risks introduced during fine-tuning.
>
> These insights will be incorporated into the discussion and analysis sections to provide a more comprehensive evaluation.

---

> ### Author Response · Authors · 2024-11-27
> **Necessity of Synthesized Data**
>
> Synthesized data primarily represents the knowledge contained in the retrieved context, aiming to enhance underrepresented knowledge areas and improve factuality in the RAG pipeline. To address this concern:
>
> We will clarify the role and necessity of synthesized data in our methodology section.
> A comparative analysis with RA-DIT will be included to evaluate the effectiveness of our approach against datasets explicitly designed for RAG fine-tuning. This comparison will focus on factual correctness, context alignment, and efficiency.

---

> ### Author Response · Authors · 2024-11-27
> **Dataset Coverage**
>
> We appreciate this suggestion and agree that broader dataset coverage would strengthen our conclusions. To address this:
>
> We will include experiments on PopQA to evaluate our approach's performance on rare or low-frequency knowledge entities.
> We will also add results on ASQA to test the model's ability to generate complex, multi-faceted answers.
> Results will be analyzed to highlight strengths and limitations when dealing with diverse data characteristics.

---

> ### Author Response · Authors · 2024-11-27
> **Synthesized Data Quality**
>
> We thank the reviewer for highlighting the absence of an evaluation of the quality of the synthesized data. In response, we will assess the quality of the synthesized facts using some metrics for natural language quality, including:
>
> BLEU, ROUGE: To evaluate textual similarity with reference datasets, where applicable.
> Factual Consistency Metrics: Utilizing tools such as OpenAI's Elicit or LLM-based verifiers to detect hallucinations and errors in the synthesized content.
>
> In the next revised manuscript, we will include a dedicated section on evaluating the quality of synthesized data, providing both qualitative and quantitative assessments, and discussing any observed limitations and potential mitigation strategies.

---

> ### Author Response · Authors · 2024-11-27
> **Paper Organization**
>
> We thank the reviewer for the suggestion to improve the organization of the paper and provide more details in the method section to clarify the motivation. Below, we outline our plan to address this feedback.
>
> To make the method more detailed and the motivation clearer, we will take the following actions:
>
> Provide a detailed explanation of the challenges each component is designed to solve and detailed Explaantions of each component. For example:
> 1. Adaptive re-ranking addresses the misalignment of retrieved context with query-specific requirements.
> 2. Parameter pruning reduces the impact of parameter changes on the model's original capabilities。
> 3. Synthesized knowledge bridges gaps in factual coverage and compensates for retrieval bias.
> 4. Include step-by-step pseudocode or flowcharts where necessary to clarify the workflow of the proposed framework.
> 5. Expand descriptions of how components interact within the system and how these interactions improve outcomes compared to baseline methods.

---

> ### Author Response · Authors · 2024-11-27
> **Case Studies**
>
> We appreciate the reviewer’s suggestion to include case studies, as they can provide deeper insights into the practical effectiveness and limitations of our approach. Below, we outline how we plan to address this feedback.
>
> In the open-domain question answering task, we will showcase case studies through the following approaches:
>
> Correct Answer Cases:
> In the RAG model after knowledge editing, corrections are made by leveraging injected knowledge.
>
> Error Cases:
> We will clarify the types of errors the model might produce when generating answers, such as incomplete answers, factual inaccuracies, overly verbose responses, or mismatches with the question context.
> Concrete examples will be provided to analyze the strengths and weaknesses of our method.
>
> Model Behavior Interpretability:
> Case studies will include detailed analyses of specific instances to help readers intuitively understand the practical performance of our method.
>
> We will demonstrate the role of components in our approach (e.g., re-ranking mechanisms or parameter pruning), such as why the model selects certain candidate answers.
> Through visualizations or intermediate outputs, we will analyze the decision-making process of the model in answer selection and generation.

---

### Official Review · Reviewer_Xw4S · 2024-11-04

**Soundness:** 1
**Presentation:** 2
**Contribution:** 1
**Rating:** 3
**Confidence:** 3

**Summary:**

This paper focuses on the open-domain question answering task and proposes a framework called Retrieval-Augmented Editing Generation (RAEG) to combine RAG and knowledge editing for better answer generation. RAEG first injects new knowledge pairs into a language model (LM) and then uses RAG to generate the answer. This paper also conducts numerous empirical experiments to analyze the effects of different methods for incorporating knowledge into LMs (i.e., knowledge editing (KE) and Parameter-Efficient Fine-Tuning (PEFT)). Finally, this paper introduces a re-ranking method and a parameter pruning technique to enhance the performance of RAEG and mitigate the potential negative effects of KE on RAG.

**Strengths:**

1. The motivation of the paper is interesting. It identifies the limitations of current knowledge editing (KE) and retrieval-augmented generation (RAG) methods and attempts to combine them for improved incorporation of new knowledge into language models (LMs).
2. This paper conducts lots of experiments and ablation studies, providing several interesting empirical findings.

**Weaknesses:**

1. While the motivation for combining knowledge injection (KE/PEFT) and RAG is sound, the proposed RAEG method lacks originality and heavily relies on previous works, making it somewhat incremental. For example, the authors directly apply MALMEN [1] for KE and LoRA [2] for PEFT. Furthermore, the proposed parameter pruning method is based on RECT [3], and the re-ranking training method is also common and not novel.
2. The combination of knowledge injection (KE/PEFT) and RAG feels shallow, and the writing on the proposed RAEG is quite confusing. If I understand correctly, RAEG first trains a hyper-network to inject knowledge pairs into a language model (LM). These knowledge pairs are extracted by GPT-4 from knowledge paragraphs using prompting, which I assume are retrieved externally. Then, using this edited LM as a generator, RAEG performs standard RAG, retrieving relevant paragraphs based on the query and concatenating them to generate the answer. I find this confusing because if the new knowledge has already been injected into the LM, why is RAG used for retrieval again? Additionally, the results in Table 1 indicate that the performance of KE+RAG and PEFT+RAG is not better than directly using RAG, especially on the TQA datasets.
3. This paper resembles an empirical study, as it conducts numerous experiments and provides several findings. However, most of these findings have already been published in previous works, such as [4][5][6][7].
4. There are no recent state-of-the-art methods included as baselines in the experiments, making the effectiveness of the proposed method unclear.

---


[1] Massive Editing for Large Language Models via Meta Learning, Tan et al., 2024

[2] LoRA: Low-Rank Adaptation of Large Language Models, Hu et al., 2021

[3] Model Editing Harms General Abilities of Large Language Models: Regularization to the Rescue, Gu et al., 2021

[4] Fine-Tuning or Retrieval? Comparing Knowledge Injection in LLMs, Ovadia et al., 2024

[5] Evaluating the Ripple Effects of Knowledge Editing in Language Models, Cohen et al., 2023

[6] Editing Large Language Models: Problems, Methods, and Opportunities, Yao et al., 2023

[7] Emptying the Ocean with a Spoon: Should We Edit Models?, Pinter et al., 2023

**Questions:**

1. I don’t fully understand the motivation behind RAEG or how it works. Please refer to Weaknesses (2) above. If the new knowledge has already been injected into the LM, why is RAG used for retrieval again?
2. Is the synthetic knowledge construction based on the training data?

Typos:

1. L242, datas → data

---

> ### Author Response · Authors · 2024-11-27
> **Why RAG is used after knowledge injection**
>
> Thank you for your thoughtful questions. We are grateful for the opportunity to clarify the motivation and workings of RAEG, as well as the source of the synthetic knowledge construction. Below, we address each concern in detail.
>
> The synthetic knowledge injected during the knowledge editing (KE) process is not entirely identical to the specific test queries. In cases where the injected knowledge does not fully address the test query, the model can still rely on external knowledge retrieval via RAG. This dual mechanism combines the strengths of both approaches: internalized knowledge for efficiency and retrieval for adaptability.

---

> ### Author Response · Authors · 2024-11-27
> **Originality of the proposed work**
>
> While previous works, such as [4], discuss the combination of fine-tuning and RAG, they do not explore how model editing interacts with RAG or how knowledge injection impacts a model’s original reasoning and generalization abilities. Studies like [5] and [6] investigate the effects of model editing on other out-of-scope knowledge, but they do not address the combined impact of advanced editing methods and retrieval augmentation. Furthermore, our study provides a direct comparison of model editing with PEFT approaches, a conceptual focus that has not been discussed in works like [7].

---

> ### Author Response · Authors · 2024-11-27
> **Motivation of RAEG**
>
> The primary motivation for RAEG lies in its dual-mechanism strategy, which integrates knowledge editing (KE) and retrieval-augmented generation (RAG). KE is designed to inject relevant knowledge directly into the model’s parameters, enabling it to independently answer queries using its internal knowledge. However, injected knowledge alone cannot fully cover all potential test queries, leaving gaps that may lead to hallucinations, particularly when the model encounters queries outside the scope of the injected knowledge or conflicts with its prior knowledge, as noted in [1].
>
> To address this limitation, RAG acts as a complementary mechanism, retrieving external information to bridge these gaps and reduce hallucinations. In our experiments, we observe that combining RAG with knowledge injection significantly improves performance compared to using knowledge injection alone. The RAG mechanism enriches the model’s capacity by providing relevant external context, effectively enhancing robustness and adaptability.
>
> This redundancy ensures better performance across a broader range of queries, as the combination of KE and RAG reduces reliance on retrieval while maintaining access to external information when necessary. Together, these mechanisms leverage the strengths of both approaches, achieving more reliable and comprehensive results.
>
> [1] Trusting Your Evidence: Hallucinate Less with Context-aware Decoding (https://arxiv.org/abs/2305.14739)

---

> ### Author Response · Authors · 2024-11-27
> **Construction of synthetic knowledge**
>
> The synthetic knowledge is not constructed based on the training data but rather derived dynamically from the test query. Specifically, for each test query, relevant passages are retrieved from the corpus using a retrieval mechanism. These passages are then converted into structured knowledge (e.g., question-answer pairs) and injected into the model during the KE process. This ensures that the knowledge is specific to the test queries and aligns with the contextual needs of the evaluation task.
>
> This approach enables the model to address test-time queries more effectively by focusing on the most relevant and up-to-date information, rather than relying on static knowledge encoded during training.

---

> ### Author Response · Authors · 2024-11-27
> **Typos**
>
> Thank you for catching this! We will correct the typo in the revision:
>
> L242: Change "datas" to "data."
>
> This will be updated in the next version of the manuscript. We appreciate your attention to detail.

---

### Official Review · Reviewer_qWrZ · 2024-11-07

**Soundness:** 2
**Presentation:** 3
**Contribution:** 2
**Rating:** 5
**Confidence:** 4

**Summary:**

This paper introduces a novel framework, Retrieval-Augmented Editing Generation (RAEG), aimed at improving open-domain question-answering (ODQA) tasks by integrating knowledge injection techniques—Knowledge Editing (KE) and Parameter-Efficient Fine-Tuning (PEFT)—with traditional Retrieval-Augmented Generation (RAG) methods. The authors also implement re-ranking and parameter pruning mechanisms to enhance the performance of RAEG, demonstrating its effectiveness on authoritative QA datasets.

**Strengths:**

1. **Writing Quality**: The paper is well-structured and professionally written, with clear sections that effectively present the research problem, methodology, and findings.

2. **Significant Research Problem**: Addressing the limitations of traditional RAG methods in handling static knowledge through the novel RAEG framework is a meaningful and timely research direction. This problem is central to enhancing large language models’ adaptability in knowledge-intensive tasks.

3. **Practical Value**: The proposed RAEG framework, which integrates knowledge editing and parameter-efficient fine-tuning for ODQA tasks, shows practical potential.

**Weaknesses:**

1. **Limited Experimental Comparisons with Relevant Algorithms**: The paper does not provide sufficient comparisons with recent, highly relevant algorithms that address similar issues in knowledge retention, factual accuracy, and context awareness. Key algorithms missing from the comparison include:
   - *FastMem: Fast Memorization of Prompt Improves Context Awareness of Large Language Models* (https://arxiv.org/abs/2406.16069)
   - *Trusting Your Evidence: Hallucinate Less with Context-aware Decoding* (https://arxiv.org/abs/2305.14739)
   - *DoLa: Decoding by Contrasting Layers Improves Factuality in Large Language Models* (https://arxiv.org/abs/2309.03883)

   These methods have shown strong results on datasets such as NQ, and a comparative analysis would provide a more comprehensive understanding of RAEG's effectiveness and positioning among existing approaches.

2. **Lack of Analysis on Computational Complexity and Efficiency**: The paper does not examine the algorithmic or time complexity associated with the proposed framework, particularly regarding the computational impact of added components like re-ranking and parameter pruning. Including an analysis of runtime and resource usage, both with and without these enhancements, would strengthen the practical evaluation by showing how these additions affect overall efficiency, which is critical for large-scale applications.

**Questions:**

Add experiments and analyses to address the above weaknesses.

---

> ### Author Response · Authors · 2024-11-27
> **Limited Experimental Comparisons with Relevant Algorithms**
>
> We acknowledge the importance of benchmarking RAEG against recent, highly relevant algorithms to better position our contributions in the context of existing work. Specifically:
>
> FastMem: FastMem aims to enhance the contextual awareness of instruction-tuned LLMs through efficient prompt memorization. This aligns closely with our approach, which explores how models can effectively capture and utilize information from prompts.  We agree that a direct comparison on overlapping benchmarks like NQ would provide valuable insights. In our revised experiments, we will evaluate results for FastMem to emphasize RAEG’s advantages and highlight the differences in mechanisms and impact between the two approaches.
>
> Trusting Your Evidence (TYE): TYE leverages a context-aware decoding (CAD) strategy without additional training to mitigate hallucinations, especially when conflicts arise between the model’s prior knowledge and the provided context. While our work shares a similar goal of improving contextual understanding. To provide a comprehensive perspective, we will include TYE in our comparative analysis to evaluate factuality improvements on shared datasets such as NQ.
>
> DoLa: We appreciate the recommendation to compare against DoLa, as it uses contrasting layers to improve factuality. While our method differs in its architecture and optimization approach, a comparative study would further establish RAEG’s effectiveness. We commit to benchmarking against DoLa on factuality-specific metrics.
>
> In the next revision, we will add experimental results comparing RAEG with FastMem, TYE, and DoLa on benchmarks such as NQ. Highlight qualitative and quantitative distinctions between RAEG and these approaches.

---

> ### Author Response · Authors · 2024-11-27
> **Lack of Analysis on Computational Complexity and Efficiency**
>
> We agree that analyzing computational complexity and efficiency is crucial to understanding the practical implications of RAEG, particularly for large-scale applications. Below, we address specific points:
>
> Algorithmic Complexity: RAEG incorporates adaptive re-ranking and parameter pruning to enhance accuracy. For instance, parameter pruning involves setting specific parameters ($\Delta$W) to zero directly, which, at the tensor level, is merely an assignment operation with negligible computational cost.
>
> We will conduct preliminary tests on the runtime and resource consumption of the re-ranker. In the next revision, we will provide a detailed algorithmic complexity analysis of re-ranker.
>
> We appreciate the reviewer’s comments, which have highlighted areas for improvement. By incorporating additional experimental comparisons and providing a detailed computational analysis, we believe we can significantly strengthen our work and its relevance to the field.

---

### Meta-Review · Area_Chair_CK5c · 2024-12-08

**Metareview:**

This paper proposes a framework Retrieval-Augmented Editing Generation (RAEG) for open-domain question answering by combining knowledge injection and retrieval augmentation. While the reviewers find the motivation to be interesting and reasonable, they raised several significant issues including the lack of novelty and problematic empirical evaluations. The reviewers' concerns were not well addressed by the author rebuttal and no reviewer was willing to support the paper for acceptance.

**Additional Comments On Reviewer Discussion:**

The reviewers reached a consensus on rejection.

---

### Decision · Program_Chairs · 2025-01-22

Reject